# A supergene underlies linked variation in color and morphology in a Holarctic songbird

Erik R. Funk [1✉], Nicholas A. Mason[2], Snæbjörn Pálsson [3], Tomáš Albrecht [4,5], Jeff A. Johnson [6] & Scott A. Taylor [1]

The genetic architecture of a phenotype can have considerable effects on the evolution of a trait or species. Characterizing genetic architecture provides insight into the complexity of a given phenotype and, potentially, the role of the phenotype in evolutionary processes like speciation. We use genome sequences to investigate the genetic basis of phenotypic variation in redpoll finches (*Acanthis* spp.). We demonstrate that variation in redpoll phenotype is broadly controlled by a ~55-Mb chromosomal inversion. Within this inversion, we find multiple candidate genes related to melanogenesis, carotenoid coloration, and bill shape, suggesting the inversion acts as a supergene controlling multiple linked traits. A latitudinal gradient in ecotype distribution suggests supergene driven variation in color and bill morphology are likely under environmental selection, maintaining supergene haplotypes as a balanced polymorphism. Our results provide a mechanism for the maintenance of ecotype variation in redpolls despite a genome largely homogenized by gene flow.

[1] Department of Ecology and Evolutionary Biology, University of Colorado, Boulder, CO 80309, USA. [2] Museum of Natural Science and Department of Biological Science, Louisiana State University, Baton Rouge, LA 70803, USA. [3] Department of Life and Environmental Sciences, University of Iceland, Askja, Sturlugata 7, 101 Reykjavik, Iceland. [4] Department of Zoology, Charles University, Vinicna 7, CZ-12844 Prague, Czech Republic. [5] Institute of Vertebrate Biology, Czech Academy of Sciences, Kvetna 8, CZ-60365 Brno, Czech Republic. [6] Wolf Creek Operating Foundation, Wolf, WY 82844, USA.
✉email: erikrfunk@gmail.com

dentifying the genetic basis of divergent traits has become a major goal in biology. Studies demonstrating associations between genotype and phenotype have provided evidence for key evolutionary processes, such as adaptive introgression[1–3] and speciation[4]. Understanding the genetic architecture of traits (e.g., single gene, polygenic, supergene) may provide insight into the complexity of the phenotype, and its role in various evolutionary processes[5]. Speciose radiations, such as the Capuchino seedeaters (*Sporophila* spp.), demonstrate that traits under independent modular genetic architecture might readily generate novel phenotypes through genetic recombination, with the potential to lead to increased species diversity[6–8]. Alternatively, traits may become tightly linked by large structural variants, such as chromosomal inversions[9,10]. When chromosomal inversions link multiple genetic elements that control a suite of traits (e.g., phenotypic, behavioral, etc.) they are commonly referred to as supergenes[11]. Because supergenes can maintain complex phenotypic differences within a population, supergenes may have unique evolutionary implications. For example, different evolutionary outcomes may depend on if (and how) incompatibilities arise between supergene haplotypes. Mutations that arise along one inversion haplotype might be locally advantageous in an isolated population, but incompatible with the alternative inversion haplotype, resulting in a classic model of lineage divergence and speciation through Bateson–Dobzhansky–Muller incompatibilities[12,13]. However, the immediate effects of an inversion (e.g., altering gene expression at inversion breakpoints) or subsequent point mutations within an inversion may result in an inversion genotype being lethal or highly deleterious to fitness, despite providing a fitness benefit to heterozygous individuals[9,10,14]. This latter model can generate a stable polymorphism that promotes intraspecific variation (such as different ecotypes) but not lineage divergence and subsequent speciation. Importantly, even in the absence of lethal genotypes, inversions may persist as stable polymorphisms if they affect traits involved in local adaptation across a heterogeneous environment[15].

Redpoll finches (*Acanthis* spp.) are high-latitude Holarctic songbirds that have a long history of taxonomic controversy due to low levels of genetic divergence and overlapping geographic distributions despite variation in body size, bill morphology, and plumage coloration (Fig. 1a; refs. [16,17]). Although these species co-occur in much of their ranges, environmental niche models demonstrate slight latitudinal differences between common and hoary redpolls during the breeding months[18]. Specifically, lighter-plumaged individuals with shorter and narrower bills (i.e., hoary redpolls) are more common at higher latitudes while lesser redpolls are restricted to western Europe and southern Scandinavia. This variation in redpoll morphology across latitude suggests phenotype may be playing a role in local adaptation; however, it is unclear the extent to which adaptation is contributing to reproductive isolation and divergence. While redpolls are commonly defined as three species, including the common redpoll (*Acanthis flammea*), the hoary redpoll (*A. hornemanni*), and the lesser redpoll (*A. cabaret*), we refer to the different redpoll groups described above as ecotypes based on phenotype and breeding distribution, given the lack of taxonomic consensus.

Here, we identify regions of the genome underlying phenotypic variation, including numerous genes inside a 55-Mb chromosomal inversion that links multiple phenotypic traits as a supergene. Genomic divergence appears constrained to these regions while the genomic background appears homogenized by ongoing gene flow. The associations identified here provide insight into the role that traits and their genetic architecture play in maintenance of phenotypic variation despite widespread gene flow.

## Results and discussion

To evaluate population structure in redpolls, we sequenced genomes of 73 individuals from the three described redpoll ecotypes (Supplementary Data File 1). Our results from whole genomes

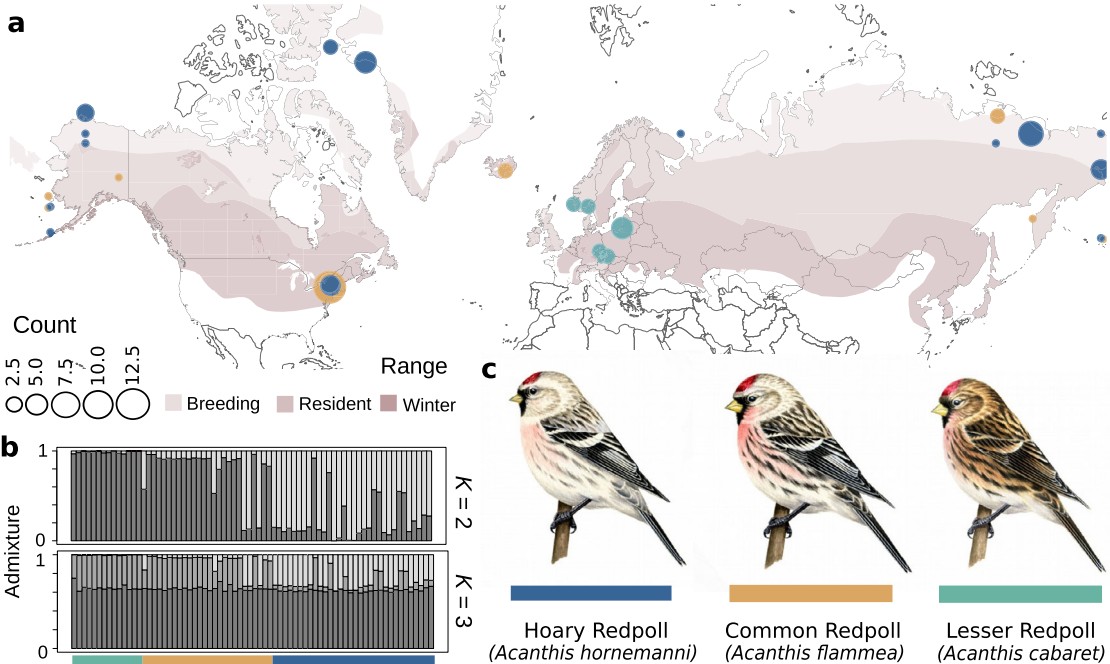

**Fig. 1 The redpoll finches. a** Seasonal distribution of redpolls shown in purple, with lightest indicating breeding range, and darkest indicating wintering range. Colored dots indicate sample number and locations with blue corresponding to hoary redpoll, gold corresponding to common redpoll, and green corresponding to lesser redpoll. **b** conStruct plots for a *K* of 2 and 3 showing that genetic clustering poorly aligns with taxonomy. **c** Illustrations of the three redpoll ecotypes demonstrating differences in size, bill, and plumage coloration (Illustrations by Liz Clayton Fuller). Distribution was generated using shape files provided by BirdLife International.

confirm findings from a previous study using a reduced-representation approach (ddRAD-seq)[18]: redpolls lack population genetic structure by either geography or ecotype boundaries (Fig. 1b), with spatially explicit clustering analyses supporting $K = 2$, and failing to group all individuals according to their species classification. In addition, principal component analysis (PCA) (Supplementary Fig. 1) of 25 million single nucleotide polymorphisms (SNPs) further reveal that PC1 explains only 3.14% of total genomic variation across all ecotypes and the majority of their global distribution. However, both PCA and population assignment analyses nonetheless indicate some degree of genetic clustering (Fig. 1b, Supplementary Fig. 1). PC1 visually separated samples into three clusters, with a left-most cluster containing both lesser and common redpolls, a right-most cluster containing almost entirely hoary redpolls, and a central cluster containing a mix of both common and hoary redpolls. However, many localities were recovered in all three groups, suggesting no influence of geography on genetic structure (Supplementary Fig. 2). Because neither geography nor ecotype were perfectly assigned to clusters, we were interested in identifying the genomic regions responsible for generating these clusters, and in investigating their potential evolutionary impacts.

**Genetics of redpoll divergence.** To identify divergent regions of the genome in redpolls, we aligned sequences to a brown-capped rosy-finch (*Leucosticte australis*) reference genome and searched for local peaks of differentiation between PCA clusters by calculating $F_{ST}$ and $d_{XY}$ in 25-kb windows across all chromosomes including all ecotypes. These scans identified a highly differentiated region across 55 Mb of chromosome 1 (Fig. 2a, c, Supplementary Fig. 3). Rerunning PCA and population assignment analyses after the removal of this chromosome either eliminated, or reduced, the variation explained (Supplementary Figs. 1 and 4), demonstrating the strong contribution of this region to total genetic differentiation in redpolls. Further, conducting a PCA using only chromosome 1 qualitatively produced much stronger definition in the three clusters originally identified (Fig. 2e). Within-group heterozygosity of the middle cluster for the highly differentiated region (0.626) was roughly double that of the outside clusters (0.388 and 0.378 for left and right clusters, respectively), suggesting that the PCA groups represent three possible inversion genotypes. We hereafter refer to these putative genotypes as AA, AB, and BB in left to right order across PC1. We do not distinguish between the ancestral and inverted haplotyes, and use the term inversion to refer to the inversion region rather than the inverted haplotype.

Broadly, the pattern of divergence recovered here is consistent with a large pericentric chromosomal inversion[10,19,20], including abrupt changes in $F_{ST}$ corresponding to the inversion breakpoints, and a central spike at the centromere (Fig. 2a, c). Reduced recombination within an inversion is expected to produce patterns of elevated linkage disequilibrium (LD), along with a decrease in nucleotide diversity along the inversion in homozygotes. These patterns are both confirmed here, including a within-cluster decrease in homozygote (AA and BB) nucleotide diversity (π), and elevated LD within the inversion when both compared to regions outside the inversion and along other chromosomes (Fig. 2c, d). To further characterize the inversion, we selected one individual each from the AA and BB genotype groups to resequence using Oxford Nanopore Technologies MinION long-read sequencing. Structural variant calling with SVIM v1.4.2[21] identified an inversion extending from 18.9 to 75 Mb along chromosome 1; however, overall number of reads supporting the variant call was low due to the size of the inversion and low yield from the MinION runs.

Because redpolls overlap extensively in distribution, species identification is made primarily on the basis of a suite of morphological characters, including plumage coloration (extent of brown and red pigments), bill size and shape, and body size. Transitioning from the AA, to AB, to BB genotype also broadly mirrors a transition in phenotype from dark to light plumage coloration, where the AA genotype is associated with dark plumage, BB is associated with light plumage, and AB is intermediate. Mason and Taylor[18] paired phenotypic measurements of plumage and bill morphology with gene expression data to reveal a strong, linear correlation between gene expression and morphology (see ref. [18], Fig. 3a). Superimposing inversion genotype on this relationship for the same individuals reveals that inversion homozygotes form the extremes of these categories, while the single heterozygote forms an intermediate (Fig. 3a). Although sample size in this comparison is small, it provides strong independent evidence that the chromosome 1 inversion plays a large role in redpoll morphology, and that phenotypic variation may be additive with respect to inversion haplotype copy number.

**Genetic associations and candidate loci.** In total, we identified 498 annotated genes within the chromosome 1 inversion region. While all genes within the inversion are likely to be linked through the suppression of recombination, and thus could be contributing equally to phenotype, we nonetheless attempted to narrow down candidate gene regions in order to infer which biological processes and pathways were potentially influenced by the inversion and identify associated regions elsewhere in the genome. To do so, we applied two approaches: (1) by compiling a list of genes that fell within the highest $F_{ST}$ peaks, and (2) by identifying SNPs significantly associated with species classification using a genome-wide efficient mixed model analysis (GEMMA)[22]. While species identity does not perfectly correlate with redpoll phenotype because they exhibit continuous phenotypic variation, the fact that species classification relies almost entirely on morphology makes it a reasonable proxy for total phenotypic variation. Finally, we annotated missense mutations within the identified genes based on a variant's location with respect to open reading frames using SNPeff v4.3[23]. Our results suggest that the vast majority of SNPs associated with phenotypic variation in redpolls are within or close to the inversion: 99% of 20,443 SNPs significantly associated with redpoll phenotype were located on chromosome 1, with only 167 located elsewhere in the genome (Supplementary Fig. 3). To evaluate the reliability of these SNPs in predicting phenotype, we used a Bayesian sparse linear mixed model in a leave-one-out cross validation framework. Predicted phenotypes suggest that allelic variation of the identified SNPs explain a significant proportion of the observed phenotypic variation ($R^2 = 0.79$; Supplementary Fig. 5).

We filtered annotations for genes that either contained, or were adjacent to, significant SNPs as identified by GEMMA or $F_{ST}$ outlier analysis, resulting in 322 genes across 7 chromosomes (Supplementary Data File 2). Within this gene set, the gene ontology category of biological regulation was overrepresented. While this category is broad and difficult to interpret meaningfully, we note that a number of genes on chromosomes 1 and 2 identified by our analysis had annotations that either relate to coloration or bird bill development or have been implicated in coloration or bird bill development in previous studies (Table 1).

Within the chromosome 1 inversion region, some of the most differentiated and significantly associated regions include key genes relating to melanin synthesis: *TYR, TYRP2, FZD4, TSKU, FSTL1*[24–29]. Both *TYR* and *TYRP2* produce melanogenic enzymes that directly synthesize melanin. In addition, *FZD4* produces a G

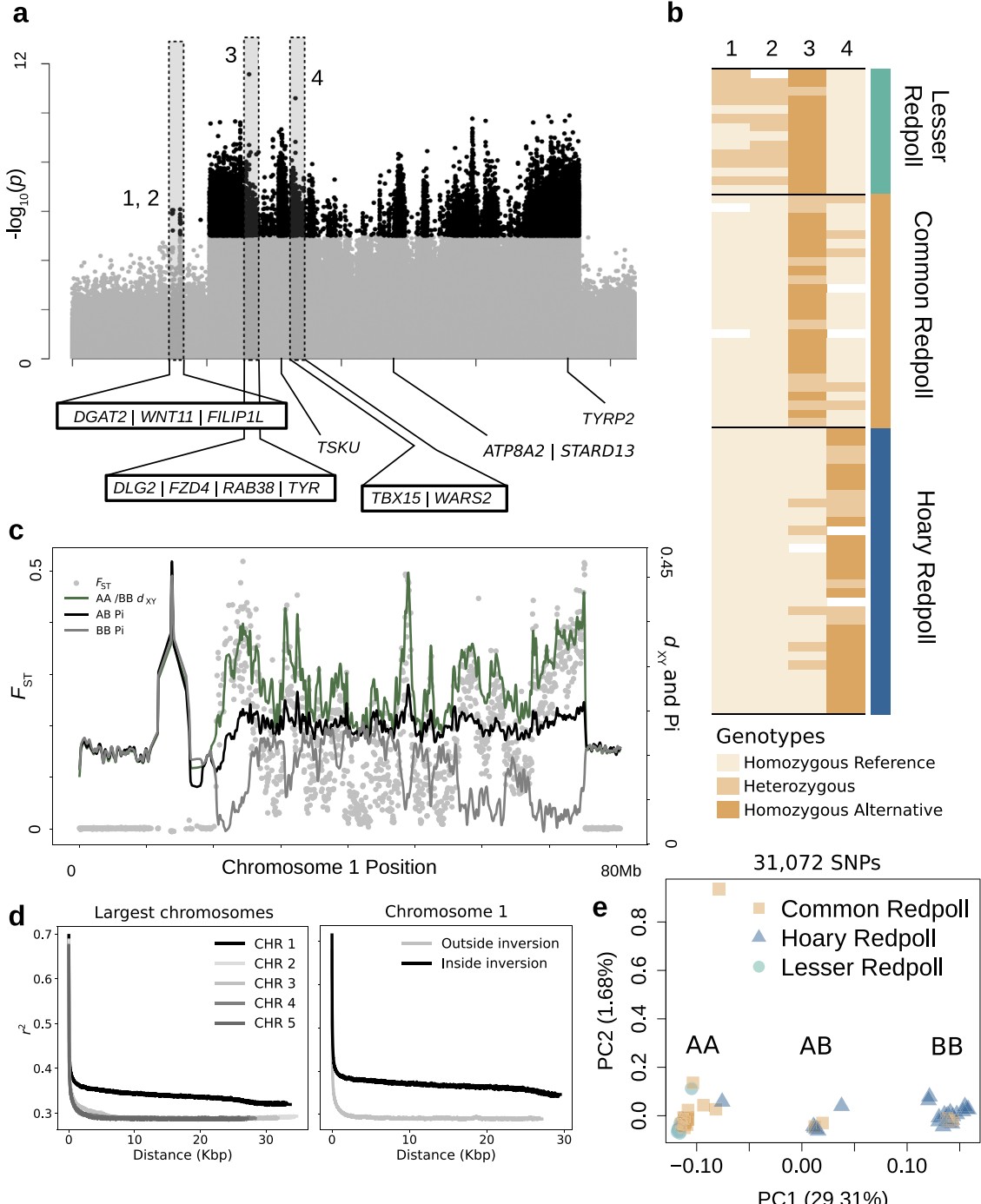

**Fig. 2 Chromosome 1 supergene. a** Chromosome 1 SNPs significantly associated with phenotype (black dots) using mixed model analysis in GEMMA with an alpha of $1 \times 10^{-5}$ to correct for multiple comparisons. Numbers indicate the two most significant SNP associations outside (1,2) and inside (3,4) the inversion region and correspond to genotypes in (**b**) for those specific SNPs. Lightest yellow cells indicate individuals homozygous for the reference allele, darkest indicate homozygous for the alternative allele, intermediate indicate heterozygotes, and white are missing data. **c** Pi (black and gray lines for AA and AB individuals, respectively) and $d_{XY}$ (green line) in 50-kb windows for the first 80 Mb of chromosome 1. **d** LD calculated as $r^2$ comparing chromosome 1 (black line) to 4 of the other largest chromosomes (shades of gray) and LD within the supergene (black) to the rest of chromosome 1 (gray). **e** PCA of chromosome 1 with a 40% minor allele frequency cutoff showing three main clusters along PC1. Taxa colors correspond to Fig. 1. All sequences were aligned to a brown-capped rosy-finch (*Leucosticte australis*) reference genome. Source data are provided as Source data file.

protein-coupled receptor in the Wingless-type signaling pathway, which acts as one of the main pathways affecting the regulation of MITF[29,30]. Previous studies of gene expression in redpolls[18] also reported differential gene expression of *FZD3*, suggesting that Frizzled family receptors may play a significant role in further modulating melanogenesis in this group.

Redpoll phenotype also varies in the amount of red feather coloration resulting from carotenoid pigmentation. Carotenoid pigments are unique in animals in that they cannot be synthesized endogenously and must instead be taken in through their diet before they can be deposited in feathers. Previous studies of genes involved in carotenoid pigmentation in birds

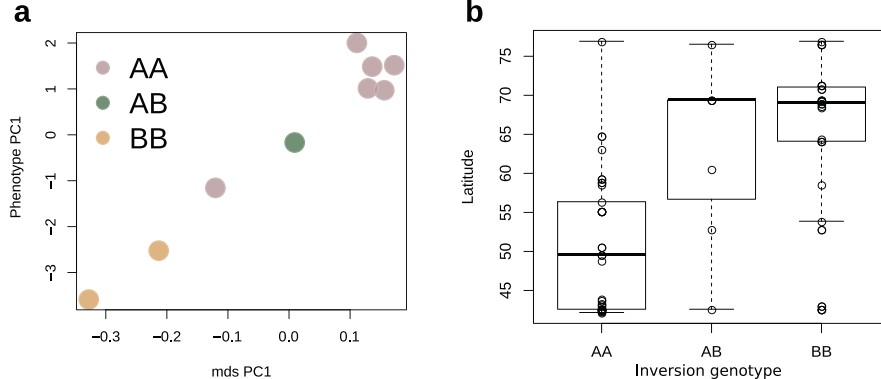

**Fig. 3 Correlations between inversion genotype, morphology, and latitude. a** Phenotype PC1 and gene expression data (as mds PC1) from Mason and Taylor (2015) colored by inversion genotype, with extreme phenotypes produced by homozygotes shown in gold and pink, and an intermediate phenotype produced by a heterozygote shown in green. **b** Latitude of sampling site for each individual grouped by inversion genotype demonstrating B haplotypes increase in frequency with latitude (AA $n = 37$, AB $n = 7$, BB $n = 28$). Box hinges represent the first and third quartiles, with centers representing medians. Whiskers represent maximum and minimum values except for BB, where outliers are values exceeding 1.5 times the interquartile range. Source data are provided as Source data file.

**Table 1 Candidate genes for plumage color and bill morphology.**

| Locus name | Location | Synonymous mutations | Missense mutations | Description |
|---|---|---|---|---|
| FILIP1L | CHR1 | 3 | 0 | Wnt pathway inhibitor |
| FSTL1 | CHR1[a] | 1 | 0 | Bone morphogenic protein inhibitor |
| TSK | CHR1[a] | 12 | 2 | Bone morphogenic protein inhibitor |
| FZD4 | CHR1[a] | 0 | 0 | Wnt pathway inhibitor |
| TYR | CHR1[a] | 17 | 1 | Catalyzes the initial step leading to melanin production |
| ATP8A2 | CHR1[a] | 14 | 8 | ATPase phospholipid transporting |
| STARD13 | CHR1[a] | 27 | 3 | StAR-related lipid transfer domain |
| TYRP2 | CHR1[a] | 24 | 7 | Regulates eumelanin and phaeomelanin |
| IL1R1 | CHR1 | 13 | 13 | SFRP4 enhancer and differentially expressed in redpolls |
| PKS15/1 | CHR2 | 67 | 37 | Possible homology with locus affecting color accumulation in Budgerigars |
| SFRP4 | CHR2 | 1 | 1 | Wnt and bone morphogenesis modulator |

[a]Located within the supergene.

highlight the role of two scavenger receptor genes (*SCARB1*[31], *SCARF2*[32]). The proteins produced by these genes likely function in the recognition of the lipoproteins that transport the hydrophobic carotenoid pigments. We identified two genes (*ATP8A2*, *STARD13*) within the inversion region that may also be related to carotenoid pigmentation through their involvement in lipid transport. Specifically, *STARD13* produces a stAR-related lipid transfer protein, which as a protein family, are involved in intracellular lipid transport, metabolism, and cell signaling events[33]. While further validation studies are required to understand the role of these genes in carotenoid variation, their functions appear to be in line with other recently reported genes associated with carotenoid pigmentation.

Two additional genes within the chromosome 1 inversion region that could be affecting phenotype are well-characterized: *TSKU* and *FSTL1* are known antagonists of bone morphogenic protein (BMP) signaling[24,26]. However, the effects of BMP inhibition may influence phenotype in at least two disparate ways: through the regulation of melanogenesis, or by contributing to differences in bill morphology. BMPs are regulators with important roles in epidermal homeostasis and hair follicle growth and pigmentation[34]. Specifically, *BMP4* and *BMP6* products have both been demonstrated as inhibiting or stimulating melanogenesis, respectively[34]. However, other studies have also implicated

*BMP4* in the development of bird bill morphology[35,36]. For example, studies of *BMP4* in Darwin's Finches find strong correlations of *BMP4* expression with both bill depth and width[35], two traits known to vary in redpolls[18,37]. Similar to Frizzled, *TSKU* was also shown to be differentially expressed in redpolls[18]. We therefore emphasize the observed differences in *TSKU* and *FSTL1* documented here could influence biologically important phenotypic variation in redpoll coloration, bill morphology, or both. Given the implication of *BMP4* in multiple pathways affecting different phenotypes, there could be pleiotropic effects resulting from one or more loci altering BMP signaling. Taken together, these candidate loci provide evidence that multiple aspects of redpoll phenotype are likely affected by a single genomic region maintaining associated SNPs from numerous genes in tight physical linkage.

While nearly all associated SNPs with gene annotations were within the chromosome 1 inversion region, three additional genes containing, or neighboring, associated SNPs may also have important phenotypic effects. Two of these—*FILIP1L* (chromosome 1 but outside of the inversion region), and *SFRP4* (chromosome 2)—act as regulators in the WNT pathway, suggesting they likely play roles in further modulating melanogenesis[38,39]. Similar to *TSKU* and *FSTL1*, *SFRP4* has also been demonstrated to regulate BMP, further emphasizing the

possibility of singular or joint effects on plumage coloration and bill morphology.

A third locus outside of the inversion region near an associated SNP on chromosome 2 includes a polyketide synthase (PKS). While this gene was annotated based on similarity to Mycobacterium *PKS15/1*, its function in birds has yet to be fully validated. However, its synteny with *RAB18*, and *YME1L1*, suggests homology with a PKS described in budgerigars (*Melopsittacus undulatus*)[40]. Functional validation through yeast-based expression demonstrated that the budgerigar PKS plays a critical role in the accumulation of red/yellow, parrot-specific pigments known as psittacofulvins. The association of PKS with redpoll phenotype indicates that it might play a similar role for organisms that contain carotenoids instead of psittacofulvins. While this requires further investigation, PKSs have been demonstrated elsewhere as important in animal pigment biosynthesis[41].

**Evolutionary consequences**. Broadly, redpoll phenotype appears to function as a balanced polymorphism resulting from a 55-Mb inversion that affects plumage coloration and bill morphology. Genetic associations that include loci outside of the inversion region suggest that phenotype is likely modulated further by several independent gene regions to generate the varied forms seen across all redpoll ecotypes. Examination of genotypes at SNPs associated with redpoll morphology (Fig. 2b) suggest that the inversion region primarily separates the hoary redpoll from both the common and lesser redpolls, while the additional associations with other genomic regions separate the lesser redpoll from both the hoary and common redpolls. These results demonstrate that the chromosome 1 inversion contains multiple, linked genetic elements that together affect a suite of phenotypic traits in redpolls, providing evidence that redpoll phenotype is broadly controlled by a supergene genetic architecture[11]. As lesser redpolls form the darkest and smallest end of the redpoll phenotype distribution, the associated SNPs located outside of the inversion may also be additive with respect to overall phenotype. Given the range restriction and more extreme phenotype of the lesser redpoll, there is less opportunity for disassortative mating, and its unclear how the derived SNPs outside of the inversion that further modulate phenotype interact with the B inversion haplotype. A previous study of an avian supergene in white-throated sparrows (*Zonotrichia albicollis*)[10] demonstrated that one of the supergene haplotypes in sparrows had likely introgressed from a closely related species. However, topology weighting across windows of the redpoll supergene favored a topology that included a sister relationship between the two haplotypes, with a combined average weight of 54% among the three topologies that included this sister relationship (Supplementary Fig. 6), providing evidence that the redpoll supergene likely evolved within the redpoll lineage[42].

Considerable theoretical attention has recently been given to the evolution and degradation of supergenes[43,44]. A primary consequence of supergene-bearing inversions is increased mutational load[12,44] due to the difficulty of purging deleterious mutations in the absence (or severe reduction) of recombination. This simple scenario could result in a balanced polymorphism stemming from associative overdominance, where inversion heterozygotes perform best because heterozygosity masks some of the deleterious mutations[12]. However, redpolls heterozygous for the inversion appear to occur in fewer numbers than homozygotes (7/73 samples in this study), suggesting an alternative mechanism may be responsible for maintaining the polymorphism. Given the presence of all three inversion genotypes in redpolls, no combination of the supergene appears to be lethal—a finding in contrast to other recently described

supergenes of similar size[9,45]. Because there is no lethal inversion genotype, recombination likely occurs regularly in homozygotes (and possibly at low levels in heterozygotes), potentially allowing for some purging of deleterious mutations. This could have a considerable influence on the maintenance of the variation, and the evolutionary consequences of this supergene.

Understanding the effects of the redpoll supergene, and the forces responsible for its maintenance, is difficult. In the absence of selection (imposed by the environment, or through mate choice), the supergene would function as a single locus with one of the haplotypes eventually becoming fixed or lost due to drift[12]. Even with some selection, high levels of migration (between inversion genotypes) would swamp out any loci contributing to local adaptation. The persistence of the redpoll supergene is therefore likely dependent on both selection and migration. One scenario that is supported by field data is that the supergene remains balanced through assortative mating. Redpolls often mate assortatively[46], but, intermediates and mixed pairs have also been observed from multiple localities[37,47]. Thus, the strictness of assortative mating may vary depending on the locality or may relax during irruptive population years[47]. Relaxation of mate choice and mixed pairings would produce the intermediate number of inversion heterozygotes seen in our data, and ultimately maintain the supergene as a stable polymorphism. However, this scenario alone does not provide an explanation for the maintenance of latitudinal differences between ecotypes. Furthermore, no hybrid zone has ever been documented in redpolls, which would be expected under a strict assortative mating scenario. While regions of hybridization have been suggested in places like Iceland, where high color variation exists[48,49], previous genetic studies have not recovered support for this hypothesis[50].

The phenotypes produced by the supergene are likely subject to environmentally mediated selection: notably, the more northerly distributed redpoll ecotype demonstrates features associated with high-latitude adaptation in other bird species (e.g., whiter color, smaller bill)[51,52]. Despite including some individuals sampled during the non-breeding season ($n = 27$), we are able to detect differences in latitude by inversion genotype group, with B haplotypes significantly more common at higher latitudes (Fig. 3b). This pattern holds when examining only breeding season birds. While it is plausible that an alternative locus is affecting ecotypic distribution, the overall low levels of background genetic variation reflect ongoing gene flow within this system. This pattern could instead reflect incomplete lineage sorting and recent divergence times, however, tests for introgression using an ABBA-BABA framework detected a significant signal of gene flow among redpoll taxa ($D = 0.0027$, $p = 0.0003$). Gene flow among ecotypes would be expected to disrupt linkage between any latitude-associated loci and phenotype through recombination unless those loci were tightly linked as in an inversion. In light of the link between the redpoll supergene and phenotype and differences in breeding distribution between ecotypes[18], the supergene may impart local adaptation to the environment. However, given the detection of inversion heterozygotes and the presence of gene flow, the inversion likely does not influence reproductive isolation. Thus, redpolls appear to function as a single species harboring ecotypic variation, rather than as three distinct species.

To explore the evolutionary conditions under which the observed pattern of the inversion polymorphism can remain balanced, we used the program SLiM[53] to simulate data under two spatial models of evolution informed by the aspects described above (Fig. 4a, b). Both models simulated 100-kb chromosomes, including a 50-kb inversion that contributed to phenotype, in diploid individuals[54]. The first model included one population with spatially varying selection along the y-axis to approximate

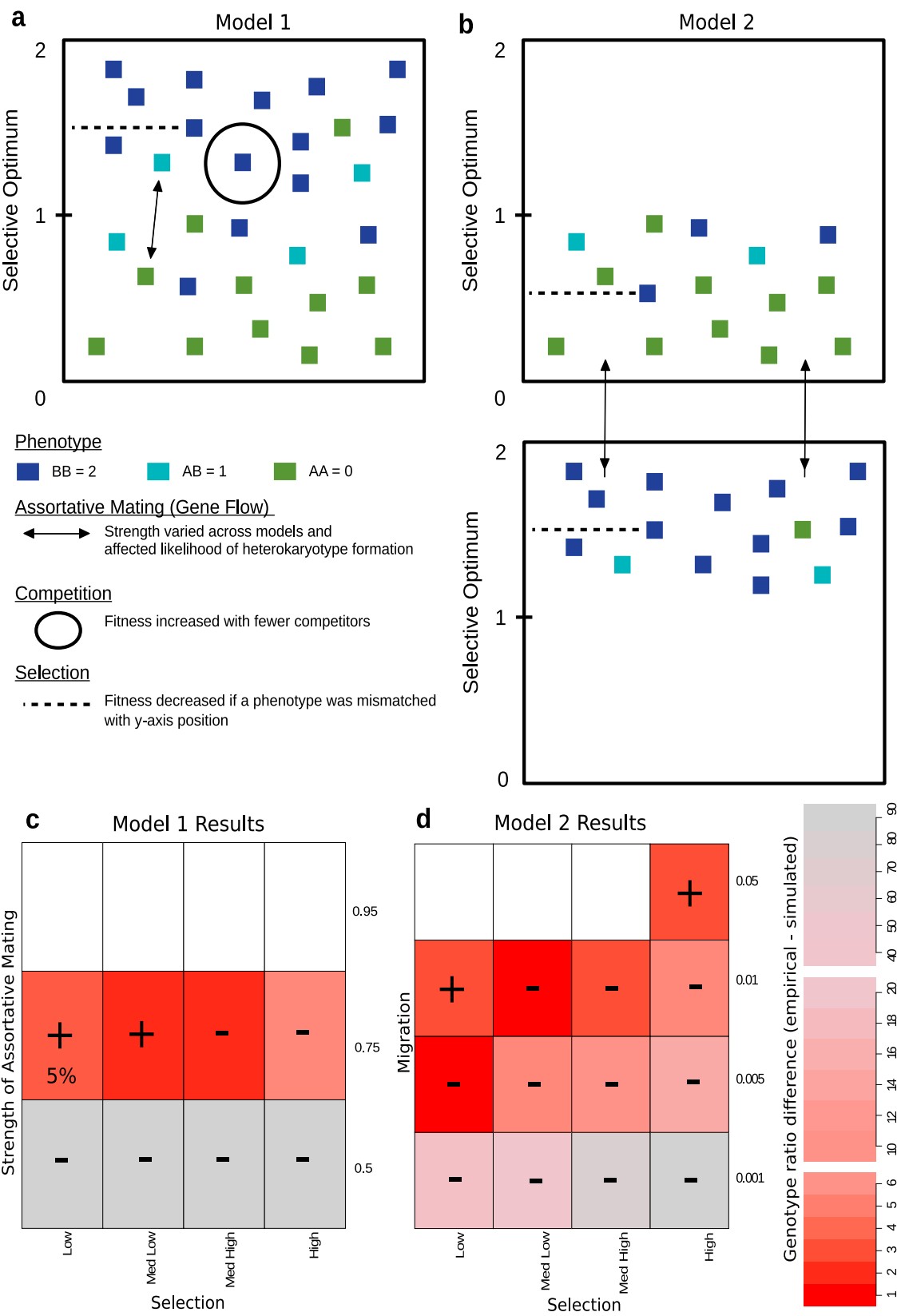

differences in fitness for a particular inversion genotype by latitude. In addition, we included assortative mating as determined by an adjustable parameter, and spatial competition such that individuals surrounded by fewer individuals in space received an increase in fitness. The second model also included spatially

varying selection but considered two ecotypes as two populations with gene flow. We then varied the strength of selection and the strength of assortative mating or migration parameters and quantified (1) whether or not a simulation resulted in a stable polymorphism, and (2) the inversion genotype ratios that were

**Fig. 4 Simulation models and results. a, b** 1000 diploid individuals were randomly placed in (x,y) coordinate space across one population in model 1 and two populations in model 2, respectively. Phenotype was additive, determined by inversion genotype, with blue squares representing BB individuals, green squares represent AA individuals, and cyan squares represent AB individuals. The strength of selection was determined for all individuals by a combination of the difference between phenotype and position along the y-axis (dotted line), and by a varied selection parameter. The strength of assortative mating or gene flow (solid arrow) was included in model 1 and model 2, respectively, and was varied across iterations. Model 1 also included a fitness adjustment for individuals based on the number of other individuals close by (solid circle). **c, d** Results from model 1 and model 2, respectively, show the similarity between the genotype ratios produced by simulation and genotype ratio recovered in our sampling of redpolls, where red represents the greatest similarity, gray represents the least similarity, and white represents simulations that failed to produce a stable polymorphism. Scale bar numbers represent the difference between the simulated and empirical genotype ratios with sign indicating either greater or fewer simulated heterozygotes. Numbers inside a cell indicate that only a portion of the 50 simulation iterations produced a stable polymorphism.

produced. We compared these ratios to the inversion genotype ratio in redpolls captured by our sampling. These simulations revealed that the strength of assortative mating, or amount of migration, played a larger role in the balancing of the inversion polymorphism than selection did at the levels tested (Fig. 4c, d; Supplementary Table 1). Regardless of the strength of selection, weak assortative mating or high migration invariably led to the loss of an inversion haplotype due to drift. Strong assortative mating or low levels of migration did maintain both haplotypes but failed to produce inversion heterozygotes. Further, all levels of selection produced spatial stratification of inversion genotypes along the selection gradient.

While these models are relatively simple and only represent two possibilities, they provide a starting point for further exploration of the complex dynamics that affect the maintenance of supergenes. For example, in redpolls, these simulations suggest that even very weak selection can produce the spatial variation seen among ecotypes, and that some relaxation of assortative mating is likely occurring, as has been proposed for populations in Canada and Alaska (USA)[37] and Norway[47].

As whole-genome sequences proliferate, an emerging body of literature is providing empirical evidence of intraspecific variation maintained through inversion polymorphism[9,10,14,15,55]. The maintenance of redpoll ecotypes via an inversion across an environmental gradient places redpolls within this growing number of species. In some cases, such as monkeyflowers (*Mimulus guttatus*)[14], inversion polymorphisms may confer sex-specific effects, and can be maintained within a population through a balance of positive and negative fitness interactions. In other cases, though not exclusive to sex-specific effects, inversions may affect phenotypes related to local adaptation, and species distributed across a heterogeneous environment may retain an inversion polymorphism through spatially varying selection, as suggested here for redpolls. This has recently been demonstrated in seaweed flies (*Coelopa frigida*)[15] and deer mice (*Peromyscus maniculatus*)[55]. In addition, studies of *Drosophila* have reported clinal variation in multiple survival traits controlled by an inversion as a result of spatially varying selection across latitude[56]. These findings in *Drosophila* highlight the need for further investigation into the selection pressures and fitness effects of the inversion we report in redpolls.

We provide evidence from whole-genome sequence data of loci associated with redpoll plumage coloration and bill morphology contained within a ~55-Mb inversion supergene. While some authorities classify redpolls as three separate species (e.g., ref. [57]), we find no evidence of genome-wide population genetic structure consistent with current taxonomy. Instead, we provide evidence that the suite of morphological traits used to describe redpoll species differences are linked within the identified supergene. The presence of all possible inversion genotypes suggests there are no lethal supergene combinations and indicate that while these traits are likely involved in local adaptation, they are not involved in reproductive isolation. Though breeding distributions vary latitudinally, even minor levels of contemporary gene flow within

broad areas of sympatry likely maintain these traits as stable polymorphisms. Manipulations involving common garden experiments, or aviary crosses will help elucidate the strength of selection and may reveal additional unknown genetic interactions with the supergene that are affecting the evolution of redpolls.

With the explosive growth in the number of sequenced genomes and increasing sophistication of analytical tools, detecting structural variants or complex genetic architectures is likely to become common. The large size and high gene content of these classes of variants may in some cases translate to large evolutionary effects. While key theoretical work continues to emerge, the further exploration of empirical patterns between phenotype and supergenes will provide useful insight into the evolutionary effects of similar genetic architectures for a wide range of organisms.

## Methods

**Sampling**. We sampled 73 individuals from across the three redpoll species currently recognized by many authorities (e.g., ref. [57]), including common redpoll ($n = 26$), hoary redpoll ($n = 33$), and lesser redpoll ($n = 14$, Fig. 1, Supplementary Data File 1) (Fig. 1 range map generated using ref. [58]). We extracted genomic DNA using a salt extraction protocol. Samples were first lysed using a homogenizing solution (0.4 M NaCl, 10 mM Tris–HCl pH 8.0, and 2 mM EDTA pH 8.0, Proteinase K), and a 20% SDS solution. We added 2 µl of glycoblue dye to aid in the identification of the DNA pellet, and precipitated DNA using a 6 M NaCl solution and 100% EtOH. DNA pellets were resuspended in 100 µl of TE buffer (10 mM Tris, 1 mM EDTA at pH 8–9). We prepared genomic libraries using the Nextera XT kit with half-reaction volumes. We pooled all 73 individuals, and sequenced whole genomes using two lanes of an S4 flow cell on an Illumina Novaseq (Illumina Inc., CA, USA). The collection and handling of all samples were done with approval and in accordance with the ethical guidelines set out by the University of Colorado Boulder Institutional Animal Care and Use Committee, the University of Iceland, Reykjavik Institution of Life and Environmental Science, the Czech Academy of Sciences Institute of Vertebrate Biology, the Greenland Home Rule Government, Danish Polar Center, and the Cornell University Institutional Animal Care and Use Committee.

Raw reads were trimmed using TrimmomaticPE[59] and aligned to a brown-capped rosy-finch (*Leucosticte australis*) reference genome using the BWA mem algorithm with default settings[60]. We called variants using bcftools mpileup[61], and filtered to keep only single nucleotide polymorphisms (SNPs) with a quality score higher than 80. We dropped one individual (R97) due to its sibling status with another individual. We removed all potentially paralogous loci by filtering SNPs with a depth lower than 2x and higher than 12x coverage, removed all SNPs with a minor allele frequency lower than 5%, and generated multiple datasets based on allowed missing data using VCFtools v 0.1.16[62]. We ran subsequent analyses on datasets allowing for no missing data (100p), and 25% missing data (75p) with concordant results between both datasets. We present results from the 75p dataset, unless noted otherwise. Scripts used in our bioinformatic pipeline are publicly available on Github (https://github.com/erikrfunk/whole_genome_bioinformatics)[63].

In addition, we selected two individuals to generate long-read whole genomes resequenced using Oxford Nanopore Technologies MinION sequencing platform. These individuals (R29, R47) were selected as representatives of two distinct inversion genotypes within redpolls. Library preparation and sequencing were carried out at Colorado State University using the SQK-LSK-109 Ligation Sequencing Kit (Oxford Nanopore Technologies, OX, UK), across 5 flow cells. Long reads were mapped using NGMLR v0.2.7[64] with default settings. Resulting bam alignment files were used to detect structural variants with the program SVIM v1.4.2[21]. We adjusted the maximum detectable variant size to 60 Mb using the −max_sv_size argument.

**Clustering analyses**. To visualize population genomic structure, we ran principal component analyses using the R package SNPRelate v1.19.3[65]. To further assess

clustering of individuals by species designation, we ran the program conStruct v.1.0.4[66]. We assessed models for all number of populations from $K = 1$ to $K = 6$ using both spatial and non-spatial models. Model performance was evaluated using a combination of cross validation and layer contribution scores. We selected the best value of $K$ as the model that exhibited the largest gains in predictive accuracy, while maintaining observable layer contributions.

**Diversity and linkage statistics**. We assessed heterogeneity in divergence across the genomic landscape using windowed calculations of diversity statistics for variant sites, including Pi, $F_{ST}$, and $d_{XY}$. All calculations were made using the python script popgenWindows.py (https://github.com/simonhmartin/genomics_general.com). We conducted these genome scans as pairwise comparisons between named species, and distinct clusters of individuals identified in PCAs. After identification of the chromosome 1 inversion, we calculated heterozygosity of the inversion for each of the three PCA clusters using R package adegenet v.2.1.3[67].

To evaluate linkage across the chromosome 1 inversion, we calculated linkage disequilibrium (LD) as $r^2$ using plink v1.9[68]. LD was calculated between pairwise SNPs across each chromosome for the entire genome. We also calculated LD separately along sections of chromosome 1 that were outside, and inside the inversion. LD for each region was averaged by combining all calculations of $r^2$ for a given distance between SNPs, resulting in a plot of LD decay.

**Association and annotations**. We used two approaches to identify regions of the genome associated with PCA clusters and phenotypic differences: $F_{ST}$ peaks, and a genome-wide mixed model analysis using GEMMA[22]. While continuous variation exists in redpolls, species classifications are based on morphological characters, including plumage coloration, and bill size and shape. We used species classification as a summary of phenotype to test for genetic associations in GEMMA and ran analyses using sex as a covariate, and without using any covariates. No differences were recovered between these two analyses and we report the results with sex as a covariate here. We included a matrix of relatedness, generated using the -gk 1 command within GEMMA. Positions of significant SNPs identified by GEMMA, and peak regions of $F_{ST}$ were used to compile lists of genes that either encompassed or neighbored these SNPs (within 100-kb) from the annotated brown-capped rosy-finch reference genome. We numbered our chromosomes based on gene content and synteny with the zebra finch (*Taeniopygia guttata*). Significance for associated SNPs were drawn using a false discovery rate of $p < 1e-5$. To evaluate the degree to which the associated SNPs can be used to explain phenotype, we used GEMMA to performed a leave-one-out cross validation using a Bayesian sparse linear mixed model (BSLMM) and the -predict 1 argument. To generate a predicted phenotype, we systematically dropped the phenotype for a single individual, rerunning the model each time. This resulted in a separate model, and a predicted phenotype for each individual. The variance in predicted phenotype was quantified using a regression against observed phenotype.

To evaluate how the genes identified by each approach may be playing a role in the generation of divergent phenotypes, we examined known gene ontologies using Panther v.16[69]. We tested for overrepresented GO categories and examined functional classification across three GO databases (molecular function, cellular component, and biological process), as well as signaling and metabolic pathways. In addition, we compared our gene lists to the over- and under-expressed gene lists produced by Mason and Taylor[18]. Finally, we categorized each redpoll variant by comparing its chromosomal position to the gene model coordinates in our reference genome using SNPeff v4.3[23]. Based on its position with respect to gene model reading frames, variants were classified as either intergenic, upstream/downstream, intronic, synonymous, or missense mutations.

**Supergene origin and maintenance**. To explore the possibility that one of the inversion haplotypes introgressed into redpolls from a closely related species, we used phylogenetic methods to test if the two inversion haplotypes were (1) sister, indicating an origin within the redpoll lineage, or (2) not sister, indicating an origin outside of the redpoll lineage followed by introgression. We generated phylogenies using 50, 100, and 200 SNP windows along the chromosome 1 inversion using Twisst[42]. Results were congruent among the three different window sizes so we only present results from the 100 SNP window analyses. Trees were generated using 4 redpolls, 2 from each of the inversion homozygote groups, and 9 additional individuals from across 5 taxa at varying degrees of divergence from redpolls. We selected these taxa based on the most recently published phylogeny for the family Fringillidae[70]. These additional individuals included two species grouped as crossbills (*Loxia leucoptera* and *Loxia curvirostra*), two species grouped as rosy-finches (*Leucosticte atrata* and *Leucosticte tephrocotis*), and the tree was rooted using a *Fringilla coelebs* genome from the NCBI Sequence Read Archive (SRR11537170).

All sequences were aligned using the same brown-capped rosy-finch (*Leucosticte australis*) reference genome and the same bioinformatics pipeline described above for redpolls. We converted the resulting VCF file into a .geno file using the parseVCF.py python script from Simon Martin (https://github.com/simonhmartin) and generated rooted phylogenies using PhyML[71]. PhyML was run

using the Twisst script phyml_sliding_windows.py with default settings. Topology weights were calculated and visualized in R using the plot_twisst.R script.

The evolutionary processes that act on a population in order to maintain a balanced polymorphism are likely complex and numerous. As a first step in understanding the maintenance of the supergene polymorphism in redpolls, we simulated 100-Kb chromosomes with a 50-kb inversion in 1000 diploid individuals using the program SLiM[53] under two different models of evolution. Individual phenotype was determined by the number of copies of the inversion an individual possessed. Each model included two spatial dimensions, with selection varying along the y-axis. The selective optimum was determined by an individual's spatial position and implemented as a fitness adjustment based on the difference between an individual's position and their phenotype. Model 1 simulated a single population with assortative mating and spatial competition, while model 2 simulated two populations (one corresponding to each homozygote group), allowing for migration each generation. Each simulation was allowed to run for 10,000 generations and was iterated 50 times. Each simulation also varied in one of two parameters, including selection, strength of assortative mating (model 1), or amount of migration (model 2). We generated a custom output that tallied the total number of iterations that resulted in a balanced polymorphism (i.e., both haplotypes still present) and extracted the average count of each inversion genotype, along with its spatial position. Eidos code for the models we used can be found on github at https://github.com/erikrfunk/redpoll_slim_models.

**Reporting summary**. Further information on research design is available in the Nature Research Reporting Summary linked to this article.

## Data availability

The raw sequence data generated in this study have been deposited in the NCBI Sequence Read Archive (SRA) database under accession code PRJNA753137. Genomic datasets are available at the Dryad Digital Repository [https://doi.org/10.5061/dryad.q83bk3jjm]. Sequence data for *Fringilla coelebs* is available from the NCBI Sequence Read Archive (SRR11537170). The Panther Gene Ontology database is available online at [http://www.pantherdb.org/]. Source data are provided with this paper.

## Code availability

Previously published code used in this study is available on github at https://github.com/erikrfunk/whole_genome_bioinformatics [https://doi.org/10.5281/zenodo.5542029], https://github.com/simonhmartin/genomics_general, and https://github.com/simonhmartin/twisst. Eidos code written for the SLiM models used in this study is available at https://github.com/erikrfunk/redpoll_slim_models [https://doi.org/10.5281/zenodo.5542015].

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

## Acknowledgements

We would like to thank David Toews, Erica Larson, Erik Enbody, Ethan Linck, and members of the Taylor Lab for comments on previous versions of this manuscript, and Aaron Westmoreland for comments on our SLiM models. We would like to thank the Natural History Museum in Oslo, the Yale Peabody Museum, the University of Alaska Museum, the Cornell University Museum of Vertebrates, and the University of Washington Burke Museum for providing samples for this project, along with Craig Benkman for providing the crossbill genome sequences used in our topology weighting analyses. We would also like to thank Kurt Burnham and the High Arctic Institute for field support in obtaining samples in Greenland, with permits provided by the Greenland Home Rule Government. T.A. would like to thank the Czech Science Foundation (projects 15-11782S and 19-22538S) for funding and the Bird Ringing Centre at the National Museum in Prague (namely Jaroslav Cepak and Petr Klvana) for their support during the sampling of redpolls. We would like to thank the Society of Systematic

Biologists, American Ornithological Society, and the Denver Field Ornithologists for providing funding for this project.

## Author contributions

S.A.T., E.R.F., and N.A.M. conceived the study. E.R.F. performed data analysis, generated figures, and wrote the manuscript with assistance from S.A.T. N.A.M., J.J., T.A., and S.P. provided critical samples and manuscript revisions.

## Competing interests

The authors declare no competing interests.

## Additional information

**Peer review information** *Nature Communications* thanks Emily Moore, Rusty Gosner, and the other anonymous reviewer(s) for their contribution to the peer review this work. Peer reviewer reports are available.

