## [Peer Review File · Nature Communications]

Reviewers' Comments:

Reviewer #1:

Remarks to the Author:

In this manuscript the authors characterize a large autosomal inversion region that is underlying phenotypic variation in redpolls. They use whole genome sequencing to locate the inversion region and then test for genotype-phenotype association between karyotypes and redpoll ecotypes. They then describe candidate genes within the inversion region that may underly the observed phenotypic variation. Finally, they construct two evolutionary models to simulate a range of scenarios for the maintenance of the inversion polymorphism in order to identify potential processes involved in maintaining variation at the inversion supergene.

Overall, I find this a fascinating study with interesting results that will interest a broad readership. The analyses seem to have been thoroughly conducted and the manuscript is well written, so it was a pleasure to review.

I have a few general comments that need to be addressed for clarity.

1. PCA. I agree that the clustering based on the whole genome data was weak but it did seem to show some clusters, for example for the Lesser Redpoll samples particularly when you exclude the inversion chromosome (Figure S1). It struck me from Figure 1 that there seemed to be a spatial bias, i.e. all Lesser Redpoll samples came all from Europe whereas the other two ecotypes were sampled across a much larger geographic area - either in Eurasia or North America. How much does geography (sample origin) explain the genotypic variation? I would find it helpful, if such an analysis is added. Along these lines, it would also provide important context, if the (breeding) distributions of all ecotypes could be shown in Figure 1.

2. One strength of the paper is the evolutionary modelling to understand the processes that potentially maintain the variation at this inversion region. However, I really struggled to follow what was done. It took me the best part of an hour to understand the output in Table 1, I would appreciate more details to help the reader with this. For example, which parameter combination is closest to the observed data? Or which is likely? Perhaps, the authors could rather visualize the results in a meaningful way, e.g. a diagram with likely and unlikely combinations - for example similar to Fig. 5 in Merot et al. 2020 Nature Communications 11:670). I understand that there is always a limit how much detail can be explained, so I looked at the model on the github page (provided in L384) but found the documentation to be rudimentary and insufficient to follow. If space in the main text is restricted, I recommend to reduce the section "Genetic associations and candidate loci" in favour of providing more details on the evolutionary models.

3. Although I do share the authors enthusiasm for birds, I found that the manuscript seems very bird heavy. This is lamentable as inversion polymorphisms are ubiquitous and found in so many taxa. Broadening the discussion, for example, on the maintenance of the balanced polymorphism would really improve the value of the study. I highly recommend to look at the relevant studies on inversion polymorphisms in yellow monkey flowers and seaweed flies that also report clinal variation in supergene alleles and discuss the presented results in this context.

Other detailed comments

L24 The entire redpoll phenotype is unlikely to controlled by the inversion region. Rather change to "that variation in redpoll phenotypes is..."

L76-77. I would suggest to add a PCA that also tries to capture geographic variation by including spatial distribution of the sample site.

L136-138. Did you find any nonsense mutations that would suggest a degeneration of one of the haplotype?

L146-204. This section can be shortened or perhaps better captured by a table with functional details (i.e. expanding Table 1)

L153-160? Are (some of) these genes part of the same gene network?

L221. Please provide a quantification. What was the combined weight of all scenarios that had the two haplotypes as a sister relationship.

L233. The inversion allele is actually not (fully) lethal in white-throated sparrows (see Horton et al. 2013, Behavioral Genetics, 43:60-70) - only in ruffs. It is lethal in fire ants though (Wang et al. 2013, Nature, 493:664-668)

L237-238 Do you mean maintenance of the variation at this supergene?

L275-276. What exactly do you mean by "spatially varying selection" (I assume it is fitness advantage for one supergene variant by latitude) and what exactly is "spatial competition"?

L280. Which karyotype ratios do you expect to observe in redpolls and based on what?

L400. Here you mention the first time the origin of the reference genome. It is crucial information required to understand e.g. Figure 2b - that the reference is actually a different species. Hence, I suggest to introduce this much earlier, i.e. in the first part of the results section. And also add this information to the legend of Figure 2.

Table 2. Lacks crucial details. Are the "Average Karyotype Ratios" in Model 2 referring to the same ones as for Model 1? Why is no ratio for Karyotype BB provided?

Figure 2. a) Not clear how you picked "the most distinct or significant associations" - what did set these four apart from others?

Figure 2. b) Add source of the reference, i.e. that the reference is not a redpoll.

Figure 2. c) Add Pi for AA?

Reviewer #2:

Remarks to the Author:

Funk et al. set out to characterize genetic architecture to provide insight into the phenotypic history of redpolls. They found a number of candidate genes that may act as a supergene controlling multiple linked traits that acts as a mechanism to maintain ecotypic variation. Through PCA analysis they were able to identify three putative karyotypes AA, AB, and BB. Found the location of the inversion was chromosome 1. There are three closely related redpoll species, common redpoll, hoary redpoll, and the lesser redpoll. Due to a lack of taxonomic consensus the refer to the three groups as ecotypes based on phenotype and breeding differentiation. Found numerous genes in a 55-mb chromosomal inversion that links many phenotypic traits that appear to be a supergene. This results from whole genome confirm previous studies that found a lack of genetic structure between the redpoll species. PCA analysis using chromosome 1 and represent three inversion karyotypes. Homokaryotypes correlated with phenotype and latitude with B haplotype increases with latitude. The supergene distinguishes Hoary from the common and lesser redpolls, while the genomics separates the lesser from the hoary and common redpolls. Simulations provide insights into how the supergene polymorphism might be maintain in redpoll populations.

I would have liked more insights into how their data does or does not help define redpolls as three separate species or one species or a species complex that is diverging. How many of each group had each karyotype: AA, AB, or BB. How common was the heterkaryotype per ecotype? The research is interesting. I am not sure there is a global conclusion other than supergenes may be common and linked with some phenotypes and persist regardless of selection. Simulations indicated that the polymorphism of regardless of selection or weak assortative mating. This would have been nice to bring this back to help resolve the redpoll species debate rather than referencing ecotypes. I wonder if possible to conclude that supergenes might be used to resolve debates of defining species? Could the others comment on how their results might help resolve the debate? I

was looking for a more global application of supergenes.

Reviewer #3:

Remarks to the Author:

In this manuscript, Funk and colleagues use whole genome sequencing of three ecotypes of Redpoll songbirds to identify genomic regions associated with bill shape and pigmentation. Comparisons between eco-morphs reveal several loci associated with phenotypic differentiation, including a large inversion on Chromosome 1. The inversion seems to be a 'supergene', as it links several candidate genes that may be involved in pigmentation and morphology, maintaining two putatively adaptive haplotypes as a balanced polymorphism despite gene flow between groups. The authors show that this inversion likely originated within Redpolls, rather than by introgression, and use forward simulations to model the parameters (selection, assortative mating, and migration) required to maintain balanced polymorphism with spatially-varying selection.

In general, I really appreciate that the authors frame the paper around bigger picture questions about the evolutionary trajectory of species with independently evolving adaptive traits vs those with key phenotypes linked into a suite of adaptive traits via a supergene genetic architecture. The combination of genomic data analysis and evolutionary models serve as complementary methods to investigate these questions. By framing the paper in this way, they have made their contributions relevant to a broader audience of evolutionary biologists. I also like that they were able to compare their candidate loci from this paper with previously identified differentially expressed genes, though I felt like this could have been emphasized more when discussing the specific biology of candidate genes. This paper really shines when discussing particular results, but is occasionally vague when making links to larger evolutionary dynamics. The analyses are all appropriate, and in general the authors are careful about building measured arguments from multiple lines of evidence. Additionally, the figures for this paper are quite lovely and easy to understand.

General notes:

I think it is incorrect to use the term 'karyotype' and 'karyotypic' here. A karyotype is a visualization of chromosomes, and might be used if one were discussing differences in chromosome number. I recommend using the term 'haplotype' instead (see usage in the review from Schwander, Libbrecht, and Keller, 2014).

This is really small, but the figure panels are not called in order in the text (for example, figure 2b isn't called until line 211). Consider reordering the panels to match the in-text order.

Specific comments:

LN 45-47: This is confusingly worded! I would specify that when suites of traits are linked by the inversion, they are more difficult to break up through recombination at the locus. I know that you are indicating *phenotypic* divergence here, but being a little more specific will help with confusion regarding predictions of genomic divergence between supergene haplotypes.

LN 51: "unique evolutionary implications" is vague here. This is a place where you can add in ideas of ecotype differentiation connected to supergenes, which will help with both the vagueness and help connect the big picture ideas here to the specific system introduced in the next paragraph.

LN 63-65: Are these the same groups referred to as 'species' in line 58, or a subset? It is good to know that you are defining them as ecotypes, but the position of this sentence may be slightly more confusing than clarifying. I might either move this up to right after the sentence where you discuss taxonomic controversy if you feel like it really belongs in the intro, or move it to when you introduce your analysis.

LN 71: Since this paper has the methods section last, it would benefit from a slight increase in the amount of explanation of the methods. You could add a single sentence quickly summarizing the experiment, such as "To evaluate population structure in redpolls, first we sequenced the genomes of 73

individuals from three ecotypes.”

LN 72: add citation

LN 95-96: Are these values for all of chromosome 1, or just the inversion? If it is whole chromosome, what are the values for just the inversion?

LN 98: The terminology and usage throughout the paper suggests that AB is heterozygous for two inversion haplotypes, A and B. Is this the case? It took me a while to realize that the AA, AB, and BB groupings are different from the (1,2)+3, 3, and 4 region groupings you highlight in 2a and 2b. Figure 2b suggests an allelic series to me. This should be clarified in text here.

LN 105-106: An in-line list with parentheses and numbers is a bit hard to follow, consider removing numbers. Suggestion: “...when both compared to regions outside the inversion and along other chromosomes (Figure 2d).”

LN 114-116: Do you mean that inversion haplotypes are associated with your phenotype? The haplotypes were identified by sequence differences between phenotype groups, so this is what you would expect to see. It might be clearer to explicitly state “haplotype AA is associated with dark plumage, BB is associated with light plumage and AB heterozygotes are intermediate” or something similar

LN 138-141: I don’t think you can claim this—there is no way to know for certain how many genes within the inversion are contributing to phenotypic variation and how many simply have linked SNPs. The number of significant variants does not indicate number of genes, and the fact that these SNPs are in LD means that they can’t be treated as independent.

LN 157: add citation

LN 167 – 171: Are these tied to carotenoid transport? It’s a little unclear how tenuous the connection is between the scavenger proteins described in previous studies and the ones found here.

LN 182: I think this might be buried too deeply — it could be worth explicitly delving into the overlap between the previous gene expression work and this work, as it is the same system and you have already introduced the overlap. It is nice that some of the DE genes are within the intervals identified here, and it’s a real strength of the paper that you can connect your findings to these data.

LN 172 – 187: What are the chances that shifts in BMP signaling, resulting from allelic variation at a single gene in the inversion, are responsible for pleiotropic downstream effects in both color and morphology? I don’t think pleiotropy is the most parsimonious explanation, but you have just made the argument that BMP signaling can effect both traits and should address this.

LN 206: The “Evolutionary Consequences” section addresses that there are other loci identified in the scan for differentiation, but then focuses almost exclusively on the discussion of the dynamics of the supergene. Can you add in discussion of how you expect supergenes to behave within a more complex genetic architecture somewhere in this section?

LN 221: In context with the last sentence, it’s a bit ambiguous if the “this supergene” is referencing the locus in your paper, or the supergene in sparrows.

LN 234-235: You also might have very low levels of recombination between inversion haplotypes, which is evident in your data with the highest F_{st} at the edges and a dip in F_{st} in the middle (I’m wondering if the high F_{st} in the center of the inversion is from the drop in diversity in the BB haplotype)

LN 271 – 287: Does this model the locus as additive? I’m less familiar with these kinds of models and how they deal with additive phenotypes with continuous selection pressures (as you might

expect in a latitudinal gradient), but it seems like modeling the trait as additive and polygenic might be most accurate given your results and continuous phenotype.

Figure 3a y-axis typo, should be "Phenotype"

Supplemental Figure S2: is there a reason for labeling by scaffold and not just by chromosome? Unless it will make comparisons to other papers or data sets more difficult, I would suggest just numbering and ordering everything by chromosome.

Emily Moore

Thank you for these reviews. We have revised our manuscript according to the comments provided here. We have included responses to each comment below. Line numbers correspond to the tracked changes version of our manuscript.

REVIEWER COMMENTS

Reviewer #1 (Remarks to the Author):

In this manuscript the authors characterize a large autosomal inversion region that is underlying phenotypic variation in redpolls. They use whole genome sequencing to locate the inversion region and then test for genotype-phenotype association between karyotypes and redpoll ecotypes. They then describe candidate genes within the inversion region that may underly the observed phenotypic variation. Finally, they construct two evolutionary models to simulate a range of scenarios for the maintenance of the inversion polymorphism in order to identify potential processes involved in maintaining variation at the inversion supergene.

Overall, I find this a fascinating study with interesting results that will interest a broad readership. The analyses seem to have been thoroughly conducted and the manuscript is well written, so it was a pleasure to review.

I have a few general comments that need to be addressed for clarity.

1. PCA. I agree that the clustering based on the whole genome data was weak but it did seem to show some clusters, for example for the Lesser Redpoll samples particularly when you exclude the inversion chromosome (Figure S1). It struck me from Figure 1 that there seemed to be a spatial bias, i.e. all Lesser Redpoll samples came all from Europe whereas the other two ecotypes were sampled across a much larger geographic area - either in Eurasia or North America. How much does geography (sample origin) explain the genotypic variation? I would find it helpful, if such an analysis is added. Along these lines, it would also provide important context, if the (breeding) distributions of all ecotypes could be shown in Figure 1.

We have added a statement regarding the distribution of the lesser redpoll that reads:

Ln 74: "...while lesser redpolls are restricted to western Europe and southern Scandinavia."

We have also added that our clustering analyses were spatially explicit, accounting for geography, and that the clusters in our PCA are not driven by geography. These revised statements read:

Line 90: "...with spatially explicit clustering analyses supporting K=2..."

Line 98: "However, many localities were recovered in all three groups, suggesting no geographic influence on genetic structure. Because neither geography nor ecotype were perfectly assigned to clusters..."

We have also generated a supplemental PCA figure that is coded by locality to help readers visualize shared geography across karyotypes (Figure S2).

2. One strength of the paper is the evolutionary modelling to understand the processes that potentially maintain the variation at this inversion region. However, I really struggled to follow what was done. It took me the best part of an hour to understand the output in Table 1, I would appreciate more details to help the reader with this. For example, which parameter combination is closest to the observed data? Or which is likely? Perhaps, the authors could rather visualize the results in a meaningful way, e.g. a diagram with likely and unlikely combinations - for example similar to Fig. 5 in Merot et al. 2020 Nature Communications 11:670). I understand that there is always a limit how much detail can be explained, so I looked at the model on the github page (provided in L384) but found the documentation to be rudimentary and insufficient to follow. If space in the main text is restricted, I recommend to reduce the section “Genetic associations and candidate loci” in favour of providing more details on the evolutionary models.

We have added a figure that includes conceptual diagrams of our models, and a heatmap of model results that indicates likely parameter combinations. This has replaced our table in the main text, and we have moved the table to the supplemental material.

Figure 4. Simulation models and results. 1000 diploid individuals were randomly placed in (x,y) coordinate space across one population in model 1 (a), and two populations in model 2 (b). Phenotype was additive, determined by inversion karyotype, and strength of selection was determined by a combination of the difference between phenotype and position along the y-axis, and a varied selection parameter. Results from model 1 (c) and model 2 (d) show the similarity between the karyotype ratios produced by simulation and karyotype ratio recovered in our sampling of redpolls, where red represents the greatest similarity, gray represents the least similarity, and white represents simulations that failed to produce a stable polymorphism. Numbers inside a cell indicate that only a proportion of the 50 simulation iterations produced a stable polymorphism.

3. Although I do share the authors enthusiasm for birds, I found that the manuscript seems very bird heavy. This is lamentable as inversion polymorphisms are ubiquitous and found in so many taxa. Broadening the discussion, for example, on the maintenance of the balanced polymorphism would really improve the value of the study. I highly recommend to look at the relevant studies on inversion polymorphisms in yellow monkey flowers and seaweed flies that also report clinal variation in supergene alleles and discuss the presented results in this context.

Thank you for the paper suggestions. We have incorporated results from these papers and others. We have expanded both our introduction and our discussion to broaden the context of our results. These added sections start on Lns 52, and 329.

Other detailed comments

L24 The entire redpoll phenotype is unlikely to be controlled by the inversion region. Rather change to “that variation in redpoll phenotypes is...”

We have changed this sentence to read:

Ln 24: “We demonstrate that variation in redpoll phenotype is broadly...”

L76-77. I would suggest to add a PCA that also tries to capture geographic variation by including spatial distribution of the sample site.

We have generated a supplemental PCA that codes points by their locality. This has been inserted as Figure S2.

L136-138. Did you find any nonsense mutations that would suggest a degeneration of one of the haplotype?

We have included columns on table 1 that summarize both synonymous and missense mutations in the top candidate genes.

L146-204. This section can be shortened or perhaps better captured by a table with functional details (i.e. expanding Table 1)

We feel this section adds an important discussion of why the mentioned associations are relevant. As such we have decided to keep the length as is; however, we would be happy to shorten it if the overall length of the paper is too long.

L153-160? Are (some of) these genes part of the same gene network?

We have grouped genes that are part of the melanogenesis network together in our discussion. Our text reads:

Ln 175: "Within the chromosome 1 inversion, some of the most differentiated and significantly associated regions include key genes relating to melanin synthesis..."

L221. Please provide a quantification. What was the combined weight of all scenarios that had the two haplotypes as a sister relationship.

We have included the combined average weight of these topologies. This section now reads:

Ln 251: "...topology weighting across windows of the redpoll supergene favored a topology that included a sister relationship between the two haplotypes, with a combined average weight of 54% among the three topologies that included this sister relationship (Figure S5), providing evidence..."

L233. The inversion allele is actually not (fully) lethal in white-throated sparrows (see Horton et al. 2013, Behavioral Genetics, 43:60-70) - only in ruffs. It is lethal in fire ants though (Wang et al. 2013, Nature, 493:664-668)

We have adjusted this citation for clarity.

L237-238 Do you mean maintenance of the variation at this supergene?

Yes. We have revised this sentence to read:

Ln 268: "This could have a considerable influence on the maintenance of the variation, and the evolutionary consequences of this supergene."

L275-276. What exactly do you mean by "spatially varying selection" (I assume it is fitness advantage

for one supergene variant by latitude) and what exactly is “spatial competition”?

We have clarified these parameters in our main text. The description of our first model now reads:

Ln 311: “ The first model included one population with spatially varying selection along the y-axis to approximate differences in fitness for a particular karyotype by latitude. Additionally, we included assortative mating as determined by an adjustable parameter, and spatial competition such that individuals surrounded fewer individuals in space received an increase in fitness”

L280. Which karyotype ratios do you expect to observe in redpolls and based on what?

The expectation was derived from the empirical karyotypes captured by our sampling. We have clarified our original sentence to read:

Ln 318: “2) the karyotype ratios that were produced. We compared these ratios to the karyotype ratio in redpolls captured by our sampling.”

L400. Here you mention the first time the origin of the reference genome. It is crucial information required to understand e.g. Figure 2b - that the reference is actually a different species. Hence, I suggest to introduce this much earlier, i.e. in the first part of the results section. And also add this information to the legend of Figure 2.

We have added this information into the beginning of our “Genetics of redpoll divergence” section in results. This sentence now reads:

Ln 105: “To identify divergent regions of the genome in redpolls, we aligned sequences to a brown-capped rosy-finch (*Leucosticte australis*) reference genome, and searched for local peaks of...”

We have also added this information to the Figure 2 caption.

Table 2. Lacks crucial details. Are the “Average Karyotype Ratios” in Model 2 referring to the same ones as for Model 1? Why is no ratio for Karyotype BB provided?

Figure 2. a) Not clear how you picked "the most distinct or significant associations" - what did set these four apart from others?

Figure 2. b) Add source of the reference, i.e. that the reference is not a redpoll.

Figure 2. c) Add P_i for AA?

We have added a figure summarizing our modeling (Figure 4), and shifted Table 2 to the supplement as Table S3. Additionally, we have added a footnote to explain why BB ratios are not provided. Briefly, because we are averaging the ratios across multiple replicates, these numbers end up being the same. Some replicates will generate more AA individuals, and some will generate more BB individuals, just by chance alone.

Reviewer #2 (Remarks to the Author):

Funk et al. set out to characterize genetic architecture to provide insight into the phenotypic history of redpolls. They found a number of candidate genes that may act as a supergene controlling multiple linked traits that acts as a mechanism to maintain ecotypic variation. Through PCA analysis they were able to identify three putative karyotypes AA, AB, and BB. Found the location of the inversion was chromosome 1. There are three closely related redpoll species, common redpoll, hoary redpoll, and the lesser redpoll. Due to a lack of taxonomic consensus they refer to the three groups as ecotypes based on phenotype and breeding differentiation. Found numerous genes in a 55-mb chromosomal inversion that links many phenotypic traits that appear to be a supergene. This results from whole genome confirm previous studies that found a lack of genetic structure between the redpoll species. PCA analysis using chromosome 1 and represent three inversion karyotypes. Homokaryotypes correlated with phenotype and latitude with B haplotype increases with latitude. The supergene distinguishes Hoary from the common and lesser redpolls, while the genomics separates the lesser from the hoary and common redpolls. Simulations provide insights into how the supergene polymorphism might be maintained in redpoll populations.

I would have liked more insights into how their data does or does not help define redpolls as three separate species or one species or a species complex that is diverging. How many of each group had each karyotype: AA, AB, or BB. How common was the heterokaryotype per ecotype? The research is interesting. I am not sure there is a global conclusion other than supergenes may be common and linked with some phenotypes and persist regardless of selection. Simulations indicated that the polymorphism of regardless of selection or weak assortative mating. This would have been nice to bring this back to help resolve the redpoll species debate rather than referencing ecotypes. I wonder if possible to conclude that supergenes might be used to resolve debates of defining species? Could the others comment on how their results might help resolve the debate? I was looking for a more global application of supergenes.

We have revised multiple sections to better address the broader context, emphasizing that we are referencing intraspecific variation. We have also included a sentence that summarizes our interpretation of redpolls as a single species.

Ln 304: "However, given the detection of heterokaryotypes and the presence of gene flow, the inversion likely does not influence reproductive isolation. As a result, redpolls appear to function as a single species harboring ecotypic variation, rather than as three distinct species."

Reviewer #3 (Remarks to the Author):

In this manuscript, Funk and colleagues use whole genome sequencing of three ecotypes of Redpoll songbirds to identify genomic regions associated with bill shape and pigmentation. Comparisons between eco-morphs reveal several loci associated with phenotypic differentiation, including a large inversion on Chromosome 1. The inversion seems to be a 'supergene', as it links several candidate genes that may be involved in pigmentation and morphology, maintaining two putatively adaptive haplotypes as a balanced polymorphism despite gene flow between groups. The authors show that this inversion likely originated within Redpolls, rather than by introgression, and use forward simulations to model the parameters (selection, assortative mating, and migration) required to maintain balanced polymorphism with spatially-varying selection.

In general, I really appreciate that the authors frame the paper around bigger picture questions about the evolutionary trajectory of species with independently evolving adaptive traits vs those with key phenotypes linked into a suite of adaptive traits via a supergene genetic architecture. The combination of genomic data analysis and evolutionary models serve as complementary methods to investigate these questions. By framing the paper in this way, they have made their contributions relevant to a broader audience of evolutionary biologists. I also like that they were able to compare their candidate loci from this paper with previously identified differentially expressed genes, though I felt like this could have been emphasized more when discussing the specific biology of candidate genes. This paper really shines when discussing particular results, but is occasionally vague when making links to larger evolutionary dynamics. The analyses are all appropriate, and in general the authors are careful about building measured arguments from multiple lines of evidence. Additionally, the figures for this paper are quite lovely and easy to understand.

General notes:

I think it is incorrect to use the term 'karyotype' and 'karyotypic' here. A karyotype is a visualization of chromosomes, and might be used if one were discussing differences in chromosome number. I recommend using the term 'haplotype' instead (see usage in the review from Schwander, Libbrecht, and Keller, 2014).

We have used the term karyotype here to maintain consistency with the majority of the literature we have reviewed.

This is really small, but the figure panels are not called in order in the text (for example, figure 2b isn't called until line 211). Consider reordering the panels to match the in-text order.

We appreciate the suggestion here; however, given the panel shapes, we are limited in their arrangement. We have tried a number of different options and found the current arrangement to work best.

Specific comments:

LN 45-47: This is confusingly worded! I would specify that when suites of traits are linked by the inversion, they are more difficult to break up through recombination at the locus. I know that you are indicating *phenotypic* divergence here, but being a little more specific will help with confusion regarding predictions of genomic divergence between supergene haplotypes.

We have revised this paragraph to both add clarity to this sentence, and further develop the ideas in the subsequent sentence. (Lns 52-64)

LN 51: "unique evolutionary implications" is vague here. This is a place where you can add in ideas of ecotype differentiation connected to supergenes, which will help with both the vagueness and help connect the big picture ideas here to the specific system introduced in the next paragraph.

We have revised this paragraph for clarity and connection to broader context. (Lns 52-64)

LNs 63-65: Are these the same groups referred to as 'species' in line 58, or a subset? It is good to know that you are defining them as ecotypes, but the position of this sentence may be slightly more confusing than clarifying. I might either move this up to right after the sentence where you discuss taxonomic controversy if you feel like it really belongs in the intro, or move it to when you introduce your

analysis.

Because we refer to breeding distribution in our classification of ecotype, we have revised the sentence in place to try to increase clarity. This sentence now reads:

Ln 78: “Given the lack of taxonomic consensus, we refer to the different redpoll groups described above as “ecotypes” based on phenotype and breeding distribution.”

LN 71: Since this paper has the methods section last, it would benefit from a slight increase in the amount of explanation of the methods. You could add a single sentence quickly summarizing the experiment, such as “To evaluate population structure in redpolls, first we sequenced the genomes of 73 individuals from three ecotypes.”

We have included the suggested sentence that summarizes our sampling and sequencing. This sentence reads:

Ln 86: “To evaluate population structure in redpolls, we sequenced genomes of 73 individuals from the three described redpoll ecotypes.”

LN 72: add citation

We have added this citation.

LN 95-96: Are these values for all of chromosome 1, or just the inversion? If it is whole chromosome, what are the values for just the inversion?

This was for just the inversion. We have revised the sentence to clarify, and it now reads:

Ln 114: “Within group heterozygosity of the middle cluster for the inverted region...”

LN 98: The terminology and usage throughout the paper suggests that AB is heterozygous for two inversion haplotypes, A and B. Is this the case? It took me a while to realize that the AA, AB, and BB groupings are different from the (1,2)+3, 3, and 4 region groupings you highlight in 2a and 2b. Figure 2b suggests an allelic series to me. This should be clarified in text here.

That is correct; the letters refer to the inversion haplotype rather than the species classification, and the numbers simply refer to snp associations. We have added clarification to the Figure 2 caption. This caption now reads:

“Numbers indicate the two most significant SNP associations inside (1,2) and outside (3,4) the inversion and correspond to genotypes in (b) for those specific SNPs.”

LN 105-106: An in-line list with parentheses and numbers is a bit hard to follow, consider removing numbers. Suggestion: “...when both compared to regions outside the inversion and along other chromosomes (Figure 2d).”

We have adopted this suggestion.

LN 114-116: Do you mean that inversion haplotypes are associated with your phenotype? The

haplotypes were identified by sequence differences between phenotype groups, so this is what you would expect to see. It might be clearer to explicitly state “haplotype AA is associated with dark plumage, BB is associated with light plumage and AB heterozygotes are intermediate” or something similar

The haplotypes were originally identified by PCA groups, so this approach acted as a validation using phenotype scores from the 2015 study.

We have expanded our sentence to include the suggested revision. This sentence now reads:

Ln 136: “Transitioning from the AA, to AB, to BB karyotype also broadly mirrors a transition in phenotype from dark to like plumage coloration, where the AA karyotype is associated with dark plumage, BB is associated with light plumage, and AB is intermediate.

LN 138-141: I don’t think you can claim this—there is no way to know for certain how many genes within the inversion are contributing to phenotypic variation and how many simply have linked SNPs. The number of significant variants does not indicate number of genes, and the fact that these SNPs are in LD means that they can't be treated as independent.

We have revised this sentence for clarity. It now reads:

Ln 160: “Our results suggest that the vast majority of SNPs associated with phenotypic variation in redpolls...”

LN 157: add citation

We have added the appropriate citations.

LN 167 – 171: Are these tied to carotenoid transport? It’s a little unclear how tenuous the connection is between the scavenger proteins described in previous studies and the ones found here.

We highlight these here simply as candidate genes. While other studies have not reported these genes specifically, and further validation studies are required, their functions appear to be in line with other studies that have demonstrated carotenoid-related gene associations. We have added this clarification at the end of the relevant paragraph.

LN 182: I think this might be buried too deeply — it could be worth explicitly delving into the overlap between the previous gene expression work and this work, as it is the same system and you have already introduced the overlap. It is nice that some of the DE genes are within the intervals identified here, and it’s a real strength of the paper that you can connect your findings to these data.

One of the limitations of our DE data is they do not come from a controlled, common garden experiment, so while we wanted to demonstrate there is an association between DE genes, phenotype, and the intervals identified here like the inversion, we did not want to stretch our results beyond what we feel they reasonably support. Future studies will try to more precisely dissect DE genes and their role in this system.

LN 172 – 187: What are the chances that shifts in BMP signaling, resulting from allelic variation at a single gene in the inversion, are responsible for pleiotropic downstream effects in both color and

morphology? I don't think pleiotropy is the most parsimonious explanation, but you have just made the argument that BMP signaling can effect both traits and should address this.

We have revised the end of the paragraph to better emphasize this possibility. The section now reads:

Ln 209: "...could influence biologically important phenotypic variation in redpoll coloration, bill morphology, or both. Given the implication of BMP4 in multiple pathways affecting different phenotypes, there could be pleiotropic effects resulting from one or more loci altering BMP signaling.

LN 206: The "Evolutionary Consequences" section addresses that there are other loci identified in the scan for differentiation, but then focuses almost exclusively on the discussion of the dynamics of the supergene. Can you add in discussion of how you expect supergenes to behave within a more complex genetic architecture somewhere in this section?

We have expanded the first paragraph of this section to further discuss the non-inversion snps.

Ln 243: "As lesser redpolls form the darkest, smallest end of redpoll distribution, the associated SNPs located outside the inversion may also be additive with respect to overall phenotype. Given the range restriction and more extreme phenotype of the lesser redpoll, there is less opportunity for disassortative mating, and its unclear how the derived SNPs outside of the inversion that further modulate phenotype interact with the B inversion haplotype.

LN 221: In context with the last sentence, it's a bit ambiguous if the "this supergene" is referencing the locus in your paper, or the supergene in sparrows.

We have revised this sentence for clarity. It now reads:

Ln 240: "... providing evidence that the redpoll supergene likely evolved within the redpoll lineage."

LN 234-235: You also might have very low levels of recombination between inversion haplotypes, which is evident in your data with the highest F_{st} at the edges and a dip in F_{st} in the middle (I'm wondering if the high F_{st} in the center of the inversion is from the drop in diversity in the BB haplotype)

The high F_{st} in the center of the inversion appeared to be the centromere, resulting in a drop in diversity in all haplotypes. We have included the possibility of heterokaryotype recombination, revising the sentence to now read:

Ln 267: "...recombination likely occurs regularly in homokaryotypes (and possibly at low levels in heterokaryotypes), potentially allowing for..."

LN 271 – 287: Does this model the locus as additive? I'm less familiar with these kinds of models and how they deal with additive phenotypes with continuous selection pressures (as you might expect in a latitudinal gradient), but it seems like modeling the trait as additive and polygenic might be most

accurate given your results and continuous phenotype.

Both models consider the locus as additive. We have added a conceptual figure (figure 4) to outline our two models and their results.

Figure 3a y-axis typo, should be “Phenotype”

We have corrected the y-axis label.

Supplemental Figure S2: is there a reason for labeling by scaffold and not just by chromosome? Unless it will make comparisons to other papers or data sets more difficult, I would suggest just numbering and ordering everything by chromosome.

One reason is that we simply don't have all of our scaffolds mapped to chromosomes yet. Our scaffolds also aren't complete chromosomes (although many are close), and so chromosome names would have to be repeated in the case of multiple scaffolds from the same chromosome. Our hope with this plot was mostly to demonstrate the genome-wide pattern of divergence.

Emily Moore

Reviewers' Comments:

Reviewer #1:

Remarks to the Author:

The revised manuscript is much improved and I'm generally content with how my comments have been addressed.

A few (mostly) minor issues remain to be addressed, please see below.

Throughout the text:

I totally agree with the reviewer 3 that "karyotype" (e.g. "inversion haplotype") is misleading, no matter how often it was used wrongly in the literature before. I suggest to change this to "haplotype", "supergene variant" etc.

Also, please check carefully whether you mean to address the "inversion" (i.e. a specific inversion haplotype) or rather the "inversion region" (i.e. supergene region that may refer to inverted and ancestral haplotypes). These terms are not the same things!

L48-50. Sentence is not clear.

Suggested change: "For example, different evolutionary outcomes may depend on if (and how) incompatibilities arise between supergene haplotypes."

(I would broaden this to "supergene haplotype" because then it includes inverted and non-inverted haplotypes).

L110. "inverted region" comes from nowhere as you introduce the inversion only a few lines below. Moreover, it is misleading as it is strictly speaking only "inverted" for the inversion haplotypes not the original haplotype. I suggest to change "inverted region" to "highly differentiated region" here.

L171, L186, L196, L211, L213, L231, L392. Should be "inversion region".

L217. Change to "A third locus outside of the inversion region, "

L262. Correct sentence – the study organism in ref. 45 was not a bird.

L332. Change to "inversion polymorphism".

L333-340. But these mechanisms do not exclude each other! For example, in both, seaweed flies and yellow monkey flowers there is evidence for both, antagonistic pleiotropy and spatially varying selection, in contributing to the maintenance of the polymorphism. Please correct this statement.

Fig. 4. Thanks for including this figure. If I understood it right a) and b) illustrate how the models work. For a, b): why is the fitness of the BB individual decreased at the higher latitude? Should it not be rather decreased for the AA individual at high latitudes? In c) and d) It would clearer if you depict whether a simulation produced higher or lower karyotype ratios than observed. What does the color/numerical scale on the bottom right refer to?

Reviewer #2:

Remarks to the Author:

The authors have done an excellent job revising the manuscript to address reviewer concerns and suggestions. The MS is much tighter in terms of scope, as well as, providing context on a more global evolutionary context. The addition of Figure 4 was very helpful to the reader. I look forward to seeing this published.

Reviewer #3:

Remarks to the Author:

The authors have thoroughly and thoughtfully addressed my previous comments. I also particularly liked the expanded discussion that included considerations of other taxa in response to Reviewer #1. All in all, I think this paper has turned out quite nicely!

REVIEWERS' COMMENTS

Reviewer #1 (Remarks to the Author):

The revised manuscript is much improved and I'm generally content with how my comments have been addressed.

A few (mostly) minor issues remain to be addressed, please see below.

Throughout the text:

I totally agree with the reviewer 3 that "karyotype" (e.g. "inversion haplotype") is misleading, no matter how often it was used wrongly in the literature before. I suggest to change this to "haplotype", "supergene variant" etc.

We have adopted this suggestion throughout the manuscript. To distinguish between whether we are referring to a single inversion copy, or the combination of copies possessed by a single individual, we use the terms "inversion haplotype" and "inversion genotype/inversion heterozygotes/inversion homozygotes" respectively.

Also, please check carefully whether you mean to address the "inversion" (i.e. a specific inversion haplotype) or rather the "inversion region" (i.e. supergene region that may refer to inverted and ancestral haplotypes). These terms are not the same things!

We have added a sentence to provide clarity to our terms.

L122: "We do not distinguish between the ancestral and inverted haplotypes, and use the term "inversion" to refer to the inversion region rather than the inverted haplotype."

L48-50. Sentence is not clear.

Suggested change: "For example, different evolutionary outcomes may depend on if (and how) incompatibilities arise between supergene haplotypes."

(I would broaden this to "supergene haplotype" because then it includes inverted and non-inverted haplotypes).

We have adopted this suggestion. (L49)

L110. "inverted region" comes from nowhere as you introduce the inversion only a few lines below. Moreover, it is misleading as it is strictly speaking only "inverted" for the inversion haplotypes not the original haplotype. I suggest to change "inverted region" to "highly differentiated region" here.

L112: We have changed the wording to "highly differentiated region".

L171, L186, L196, L211, L213, L231, L392. Should be "inversion region".

We have changed the wording at these locations to now say "inversion region".

L217. Change to "A third locus outside of the inversion region, "

We have adopted this suggestion. (L229)

L262. Correct sentence – the study organism in ref. 45 was not a bird.

We have corrected this by removing the word “birds”. The sentence now reads:

L274: “... a finding in contrast to other recently described supergenes of similar size.”

L332. Change to “inversion polymorphism”.

We have removed the “s” so the phrase is now “inversion polymorphism” (L346)

L333-340. But these mechanisms do not exclude each other! For example, in both, seaweed flies and yellow monkey flowers there is evidence for both, antagonistic pleiotropy and spatially varying selection, in contributing to the maintenance of the polymorphism. Please correct this statement.

We have clarified this statement so that it now reads:

L350: “In other cases, though not exclusive to sex-specific effects, inversions may affect phenotypes related to local adaptation, and species distributed across a heterogenous environment may retain an inversion polymorphism through spatially varying selection, as suggested here for redpolls.”

Fig. 4. Thanks for including this figure. If I understood it right a) and b) illustrate how the models work. For a, b): why is the fitness of the BB individual decreased at the higher latitude? Should it not be rather decreased for the AA individual at high latitudes? In c) and d) It would clearer if you depict whether a simulation produced higher or lower karyotype ratios than observed. What does the color/numerical scale on the bottom right refer to?

We have coded fitness to affect all inversion genotypes, not just BB, depending on their location along the y-axis. We have included the following description in our figure legend: “...strength of selection was determined by a combination of the difference between phenotype, and position along the y-axis, an by a varied selection parameter.”

We have added + and - signs to our heatmap panel to indicate either an excess or shortage of heterozygotes in relation to our empirical genotype ratio for each set of simulation parameters. We have also clarified our scale by adding a scale axis label, and including a description in our caption that reads:

“Scale bar numbers represent the difference between the simulated and empirical genotype ratios.”

Reviewer #2 (Remarks to the Author):

The authors have done an excellent job revising the manuscript to address reviewer concerns and suggestions. The MS is much tighter in terms of scope, as well as, providing context on a more global evolutionary context. The addition of Figure 4 was very helpful to the reader. I look forward to seeing this published.

Reviewer #3 (Remarks to the Author):

The authors have thoroughly and thoughtfully addressed my previous comments. I also particularly liked the expanded discussion that included considerations of other taxa in response to Reviewer #1. All in all, I think this paper has turned out quite nicely!